# Convolutional neural networks with extra-classical receptive fields

**Brian Hu**[*]
Allen Institute for Brain Science
Seattle, WA 98109
`brianh@alleninstitute.org`

**Ramakrishnan Iyer**[*]
Allen Institute for Brain Science
Seattle, WA 98109
`rami@alleninstitute.org`

**Stefan Mihalas**
Allen Institute for Brain Science
Seattle, WA 98109
`stefanm@alleninstitute.org`

## Abstract

In the visual system, neurons respond to a patch of the input known as their classical receptive field (RF), and can be modulated by stimuli in the surround. These interactions are often mediated by lateral connections, giving rise to extra-classical RFs. We use supervised learning via backpropagation to learn feedforward connections, combined with an unsupervised learning rule to learn lateral connections between units within a convolutional neural network. These connections allow each unit to integrate information from its surround, generating extra-classical receptive fields for the units in our new proposed model (CNNEx). We demonstrate that these connections make the network more robust and achieve better performance on noisy versions of the MNIST and CIFAR-10 datasets. Although the image statistics of MNIST and CIFAR-10 differ greatly, the same unsupervised learning rule generalized to both datasets. Our framework can potentially be applied to networks trained on other tasks, with the learned lateral connections aiding the computations implemented by feedforward connections when the input is unreliable.

## 1   Introduction

While feedforward convolutional neural networks have resulted in many practical successes [1], they are highly susceptible to adversarial attacks [2]. In contrast, the brain makes use of extensive recurrent connections, including lateral and feedback connections, which may provide some level of immunity to these attacks (for results on human adversarial examples, see [3]). Additionally, the brain is able to build rich internal representations of information with little to no labeled data, which is a form of unsupervised learning, in contrast to the supervised learning required by most models.

We present a model incorporating lateral connections (learned using a modified Hebbian rule) into convolutional neural networks, with feedforward connections trained in a supervised manner. When applying different noise perturbations to the MNIST [4] and CIFAR-10 [5] datasets, lateral connections in our model improve the overall performance and robustness of these networks. Our results suggest that integration of lateral connections into convolutional neural networks is an important area of future research.

---

[*]Equal contribution.

Real Neurons and Hidden Units Workshop at the 33rd Conference on Neural Information Processing Systems (NeurIPS 2019), Vancouver, Canada.

**(A)** Experiment 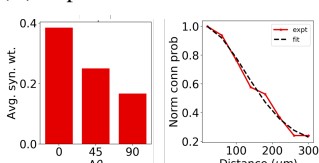 **(B)** Model positive wts 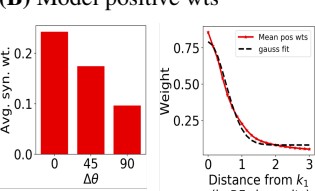 **(C)** Model negative wts 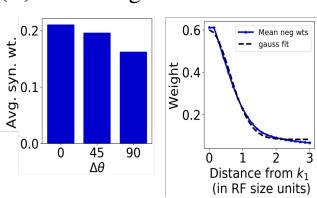

Figure 1: Orientation and distance dependence of lateral connections. **A)** Left: Connection probability as a function of difference in preferred orientation between excitatory neurons observed experimentally (from [6]). Right: Normalized connection probability between excitatory neurons as a function of inter-somatic distance as reported experimentally in mouse auditory cortex [7]. **B, C):** Model predictions for orientation and distance dependence ($k_1$ represents the target neuron) of positive (B) and negative (C) lateral connection weights for filters constructed using estimates of spatial receptive field (RF) sizes from in-vivo recordings in mouse V1 [8]. Red (blue) bars/lines represent positive (negative) weights and dashed black lines represent Gaussian fits for distance dependence (standard deviations $\sigma_{expt} = 114\mu m$, $\sigma_{pos} = 120\ \mu m$ and $\sigma_{neg} = 143\ \mu m$ for experiment, model positive and negative weights respectively). Predicted connections qualitatively match with experimental data.

**Relation to previous work**

A number of normative and dynamical models relating contextual modulation of neuronal responses and lateral connectivity have been proposed in the literature. Normative models based on sparse coding [9, 10, 11, 12, 13, 14, 15] predict anti-Hebbian lateral connections between excitatory neurons, in contrast with the experimentally observed like-to-like excitatory connectivity (but see [16] which extends the sparse coding model to learn like-to-like horizontal connections by including a pairwise coupling term in the prior).

Our model bears close resemblance to the MGSM (mixture of Gaussian scale mixtures) model of natural images proposed by Coen-Cagli, Dayan and Schwartz [17], which infers contextual interactions between the RF and surround that would lead to optimal coding of images. Recent work has demonstrated the ability to learn either flexible normalization [18] or divisive normalization [19] in deep neural networks. In addition to these approaches, other normalization schemes from the machine learning field (e.g. batch normalization [20], layer normalization [21], local response normalization [22]) have been used to accelerate training of neural networks.

In contrast with these approaches, we propose a computational role for each pyramidal neuron (or unit in a deep neural network) in terms of how it integrates lateral input information optimally from a Bayesian perspective. Our proposal enables us to incorporate optimal lateral connections (learned in an unsupervised manner) straightforwardly into feedforward neural networks.

## 2   Methods

**Neuroscience-inspired extra-classical receptive field model.** Let $fc_{j,x}^{m,(l)}$ denote the classical receptive field (RF) response of a unit representing feature $j$ in layer $l$ at image location $m$, given image $x$. The full response of a unit representing feature $j$ at image location $m$ in layer $l$, given image $x$ can then be written as:

$$f_{j,x}^{m,(l)} = fc_{j,x}^{m,(l)}\Big(1 + \alpha \sum_k \sum_{n \neq m} W_{jk}^{mn,(l)} fc_{k,x}^{n,(l)}\Big) \tag{1}$$

where the second term on the right side represents the contribution from the extra-classical RF, $\alpha$ represents a hyperparameter that tunes the strength of the lateral connections, and $W_{jk}^{mn,(l)}$ are the synaptic weights from surrounding units $n$ on to unit $m$ within layer $l$. These weights are learned in an unsupervised manner using the rule:

$$W_{jk}^{mn,(l)} = \frac{\left\langle fc_j^{m,(l)} fc_k^{n,(l)} \right\rangle_x}{\left\langle fc_j^{m,(l)} \right\rangle_x \left\langle fc_k^{n,(l)} \right\rangle_x} - 1 \tag{2}$$

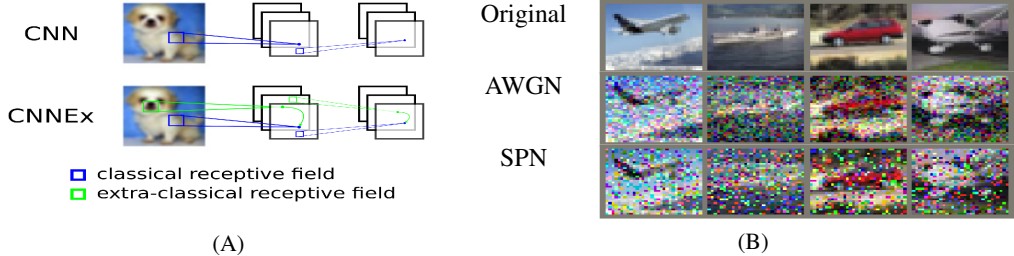

(A)               (B)

Figure 2: Overview of models and experiments. (A) Network architectures with feedforward classical receptive fields (RFs) (blue) and extra-classical RFs (green), termed CNN (top) and CNNEx (bottom), respectively. Extra-classical RFs modulate activity of other units within the network via lateral connections learned in an unsupervised manner. (B) Example images used in the experiments. Top row: original images. Middle row: AWGN stimuli. Bottom row: SPN stimuli. AWGN stimuli shown here had a mean of zero and a standard deviation of 0.2. SPN stimuli here had a fraction of changed pixels set to 0.2. Original images are reproduced from the CIFAR-10 [5] dataset.

where $\langle . \rangle_x$ represents an average over the set of training images. Note that this formula differs from a Hebbian learning rule, in that only the covariance between the feedforward responses of units leads to changes in the lateral connections. A derivation of the above equations is shown in the Appendix.

**Comparison to experimental data.** Using natural images from the Berkeley Segmentation Dataset [23] and a dictionary of classical RF features parameterized from mouse V1 electrophysiological responses [8], we computed lateral connection weights using Eq. 2. Figure 1 shows that predicted lateral connections qualitatively match the orientation and distance dependence of connectivity observed in mouse cortex.

**Datasets.** We used the standard train/test splits, holding out 10% of the training data for validation. We also added two types of noise to the original images: additive white Gaussian noise (AWGN) and salt-and-pepper noise (SPN). The mean of the AWGN was set to zero and the standard deviation varied in increasing levels from 0.1-0.5. For the SPN, the fraction of noisy pixels varied in increasing levels from 0.1-0.5. Example stimuli (original and noisy images) are shown in Figure 2.

**Network architecture and training.** The model architectures used are described in Table 1. The CNNEx and CNN-PM models have the same number of parameters, with the additional convolutional layers in CNN-PM being standard feedforward layers trained in a supervised manner (in contrast to the lateral connections given by Eq. 1 for CNNEx, which are trained in an unsupervised manner). MNIST models were trained for 10 epochs with a minibatch size of 64 using stochastic gradient descent with a learning rate of 0.01 and a momentum value of 0.5. CIFAR models were trained for 50 epochs and a momentum value of 0.9, keeping all other hyperparameters the same. We trained 10 different instantiations using different random seeds to ensure the robustness of our results. All experiments were performed using Pytorch (v. 0.3.1) on a NVIDIA GTX 1080 Ti GPU.

|  | **Model** | **Network architecture** | | | | | | | |
|---|---|---|---|---|---|---|---|---|---|
| MNIST | CNN | conv5-10 | | conv5-20 | | FC-50 | | FC-10 | soft-max |
|  | CNNEx/CNN-PM | conv5-10 | *conv7-10* | conv5-20 | *conv3-20* | FC-50 | | FC-10 | soft-max |
| CIFAR-10 | CNN | conv5-96 | | conv5-128 | | conv5-256 | FC-2048 | FC-2048 | FC-10 | soft-max |
|  | CNNEx/CNN-PM | conv5-96 | *conv7-96* | conv5-128 | *conv3-128* | conv5-256 | FC-2048 | FC-2048 | FC-10 | soft-max |

Table 1: Model architectures used. CNN is the baseline model, CNNEx is the model with optimal lateral connections, and CNN-PM is a parameter-matched model. Convolutional layers are denoted as "conv<receptive field size>-<number of channels>". Convolutional layers in italics represent lateral connections learned in an unsupervised manner. "FC" denotes fully connected layers with the given number of units. Max pooling with a 2x2 window and a stride of 2 was used after each set of convolutional layers (not shown). The ReLU activation function is also not shown for brevity.

| | Models | Original | AWGN | | | | | SPN | | | | |
|---|---|---|---|---|---|---|---|---|---|---|---|---|
| | | - | 0.1 | 0.2 | 0.3 | 0.4 | 0.5 | 0.1 | 0.2 | 0.3 | 0.4 | 0.5 |
| MNIST | CNN | 98.71 | 98.61 | 98.21 | 96.88 | 92.03 | 81.78 | 97.28 | 92.01 | 80.85 | 65.29 | 48.28 |
| | CNNEx | 97.25 | 97.17 | 96.83 | 95.86 | 93.34 | **88.24** | 96.06 | 93.45 | **87.97** | **77.99** | **63.04** |
| | CNN-PM | **98.76** | **98.68** | **98.39** | **97.49** | **94.56** | 87.92 | **97.72** | **94.66** | 87.45 | 75.17 | 59.26 |
| | CNN (wd+d) | 97.39 | 97.30 | 97.01 | 96.43 | 95.23 | 92.82 | 96.63 | 95.16 | 92.01 | 85.61 | 73.98 |
| | CNNEx (wd+d) | 97.17 | 97.10 | 96.84 | 96.32 | 95.35 | **93.59** | 96.49 | 95.30 | **93.11** | **88.53** | **79.30** |
| | CNN-PM (wd+d) | **98.14** | **98.06** | **97.80** | **97.17** | **95.50** | 91.71 | **97.34** | **95.61** | 91.53 | 83.48 | 70.07 |
| CIFAR-10 | CNN | **78.86** | **69.65** | 42.72 | 23.83 | 16.14 | 13.30 | 44.51 | 22.98 | 15.51 | 12.84 | 11.54 |
| | CNNEx | 65.66 | 64.08 | **53.47** | **37.77** | **24.37** | **16.95** | **55.24** | **37.91** | **23.22** | **15.36** | 12.22 |
| | CNN-PM | 78.44 | 68.42 | 45.77 | 27.84 | 18.97 | 15.11 | 47.07 | 26.86 | 18.24 | 14.46 | **12.72** |
| | CNN (wd+d) | 72.29 | 58.91 | 34.69 | 22.10 | 17.47 | 15.82 | 36.16 | 21.30 | 17.14 | 15.50 | 14.46 |
| | CNNEx (wd+d) | 63.24 | **59.43** | **49.96** | **40.20** | **31.45** | **24.70** | **50.64** | **39.63** | **30.13** | **22.59** | **17.73** |
| | CNN-PM (wd+d) | **77.07** | 57.39 | 28.74 | 15.77 | 12.03 | 10.92 | 30.11 | 15.43 | 11.92 | 10.79 | 10.42 |

Table 2: Model accuracy (%) on the MNIST and CIFAR-10 datasets. We separate results for the original images and the two types of noise perturbations by columns (AWGN: additive white gaussian noise, SPN: salt-and-pepper noise). The results for the baseline model (CNN), the model with lateral connections (CNNEx), and the parameter-matched model (CNN-PM) are shown in separate rows. $wd$ corresponds to models trained with weight decay = 0.005. $d$ corresponds to models trained with a dropout fraction of 0.5. All reported values are averages over 10 random initializations.

**Lateral connections.** We first trained feedforward weights in the network using supervised learning. After freezing the feedforward weights, we introduced lateral connections given by Eqn. 2 between units in the first two convolutional layers. Importantly, we note that the lateral connection weights are only learned once at the end of supervised training, by keeping the feedforward weights of the network fixed and using the activations of units over a set of training images. We do not update the feedforward weights of the network by backpropagating through the computed lateral connections. Future work will explore new methods for semi-supervised learning, which combine supervised learning of feedforward weights with unsupervised learning of the lateral connections.

**Network regularization.** We chose a weight decay value of 0.005 and a dropout fraction of 0.5. Weight decay acted on all non-bias parameters of the model, while dropout was applied after each convolutional layer in the model, as well as after the first fully connected layer. We also tested the combination of these regularization techniques with the lateral connections.

**Validation and testing.** Lateral connections had a spatial extent of 7x7 (3x3) pixels in the first (second) convolutional layers, with connections from the same spatial location set to zero. Hyperparameters $\alpha$ for each of the two layers were chosen based on a grid search over the parameter range $\{0.1, 0.01, 0.001, 0.0001\}$ using the validation dataset, followed by a finer search starting from this coarse value. We did not use lateral connections for the two fully-connected layers.

## 3 Results

**Model performance.** Table 2 summarizes model test accuracy with and without regularization (weight decay + dropout) for the network architectures described in Table 1. For MNIST, the learned lateral connections provide improvement over the baseline and parameter-matched models only at higher levels of Gaussian noise (0.5) and salt-and-pepper noise (0.3 and above). For CIFAR-10, we found that the learned lateral connections provide improved robustness to noise for almost all noise levels, with the cost of slightly decreased accuracy on the original images. We did not try fine-tuning the models after incorporating the learned lateral connections, which may help recover some of this loss in accuracy. Furthermore, we found that lateral connections combined with regularization often resulted in even better performance on both MNIST and CIFAR-10.

**Effect of lateral connections.** Lateral connections had two complimentary roles: redundancy reduction leading to sparsification of feature activations on images without noise, and noise reduction on noisy images leading to feature activations closer to those on the original images. The use of contextual information to modulate unit responses may underlie the ability of the CNNEx model to achieve higher accuracies under noisy conditions. We show both effects on features in the first convolutional layer of our MNIST model (Figure 3).

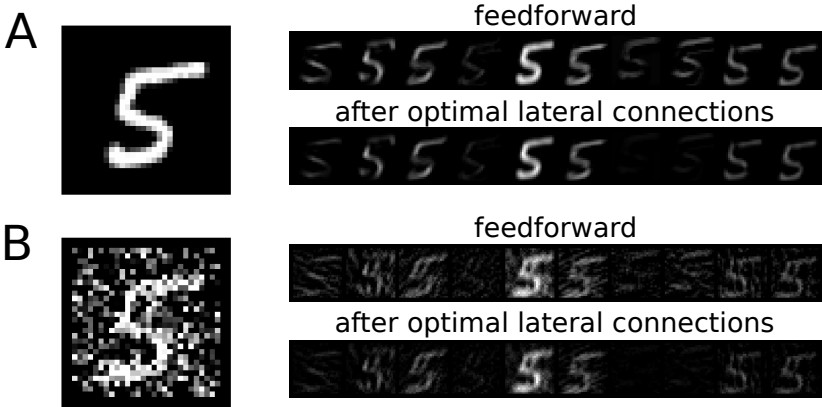

Figure 3: Effect of optimal lateral connections on feature activations in the first convolutional layer of CNNEx trained on MNIST. The first column shows the input image, while the other columns show the activations of the ten different features in the first convolutional layer before and after optimal lateral connections are applied. (A) Original image. The optimal lateral connections reduce some feature activations (e.g. features in columns 7 and 8), resulting in a sparser population code. (B) Noisy image. The optimal lateral connections reduce some feature activations associated with noise.

## 4   Discussion

In our model, lateral connections capture structure in the statistics of the world via unsupervised learning. This structure allows for inference that can make use of the integration of information across space and features. By combining these lateral connections with features learned in a supervised manner using backpropagation, the network does not learn any arbitrary structure present in the world, but only the structure of features which is needed to solve a particular task. As a result, our method allows us to predict the structure of the world which is relevant to a given task.

The vast majority of deep neural networks are feedforward in nature, although recurrent connections have been added to convolutional neural networks [24, 25]. Recurrent connections have also been used to implement different visual attention mechanisms [26, 27]. However, these networks are still largely trained in a supervised manner. An exception are ladder networks, which have been proposed as a means to combine supervised and unsupervised learning in deep neural networks [28]. However, different from our approach, ladder networks use noise injection to introduce an unsupervised cost function based on reconstruction of the internal activity of the network. Our model instead relies on a modified Hebbian learning rule which learns the optimal lateral connections between features within each layer based solely on the activations of units coding for these features.

Neurons are inherently noisy, and their responses can vary even to the same stimulus. These neurons are embedded in cortical circuits that must perform computations in the absence of information, such as under visual occlusion. Optimal lateral connections can provide additional robustness to these networks by allowing integration of information from multiple sources (i.e. different features and spatial locations). This type of computation is also potentially useful for applications in which artificial neurons are not simulated with high fidelity, e.g. in neuromorphic computing.

We chose a relatively simple network architecture as a proof-of-concept for our model. As such, we did not achieve state-of-the art performance on either image dataset. This accuracy could be further improved by either fine-tuning models after learning the optimal lateral connections or using deeper network architectures with more parameters. Future experiments will also have to test the scalability of learning optimal lateral connections on more complex network architectures and larger image datasets (e.g. ImageNet), and whether these connections provide any benefit against noise or other types of perturbations such as adversarial images.

## Acknowledgements

We wish to thank the Allen Institute founder, Paul G. Allen, for his vision, encouragement, and support.

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

## Appendix

### The model

We assume a simple neural code for each excitatory neuron: the steady-state firing rate of the neuron maps monotonically to the probability of the feature that the neuron codes for being present in the image (similar to codes assumed in previous studies [29, 30, 31]). We have

$$f_{k,x}^n = g\left(p(F_k^n | im_x)\right) \qquad (3)$$

where $f_{k,x}^n$ represents the firing rate of a neuron coding for feature $F_k$ at location $n$ in image $x$, $p(F_k^n | im_x)$ represents the probability of presence of the corresponding feature and $g$ is a monotonic

function. For simplicity, we assume a linear mapping between the probability of feature presence and firing rate ($g(y) = y$) in the rest of the paper, as the qualitative conclusions are not dependent on this choice. We subdivide the image into multiple patches corresponding to the size of the classical RF. We define the classical RF response of the neuron (with $g(y) = y$) as

$$fc_{k,x}^n = p(F_k^n | im_x^n) \qquad (4)$$

where $im_x^n$ denotes the image patch at location $n$. We require that the sum of probabilities of all features in a patch is one, for every image, thereby implying a normalization of classical RF responses in a spatial region equal to the size of a patch so that [2]

$$\sum_k fc_{k,x}^n = 1 \quad \forall n, x \qquad (5)$$

We show that a network of neurons can directly implement Bayes rule to optimally integrate information from the surround (see Supplementary Information (SI) for the derivation). Intuitively, the activity of a neuron representing a feature is influenced by the probability that another feature is present in a surrounding patch and by the statistics of co-occurrence of these features. In such a network, the activity of a neuron representing feature $j$ in patch $m$, given image $x$, can be shown to be (see SI)

$$f_{j,x}^m = \frac{1}{\mathcal{N}_x^m} fc_{j,x}^m \prod_{n \neq m}^N \left( 1 + \sum_k W_{jk}^{mn} fc_{k,x}^n \right) \qquad (6)$$

In Eq. (6), $\mathcal{N}_x^m$ represents a normalization coefficient (see SI). The term $W_{jk}^{mn}$ represents a weight from the neuron coding for feature $k$ in patch $n$ to the neuron coding for feature $j$ in patch $m$ and can be estimated as:

$$W_{jk}^{mn} = \frac{\left\langle fc_{j,x}^m fc_{k,x}^n \right\rangle_x}{\left\langle fc_{j,x}^m \right\rangle_x \left\langle fc_{k,x}^n \right\rangle_x} - 1 \qquad (7)$$

where $x$ spans the set of images used and $\langle . \rangle_x$ represents the average over all images in the set. Thus, lateral connections between neurons in our network are proportional to the relative probability of feature co-occurrences above chance in the set of images used. While the formalism can be applied to any scene statistics, we focus here on the analysis of natural scenes. Eq. (6) encapsulates a local computation by a network of excitatory neurons - optimal context integration - through *functional* lateral connections given by Eq. (7).

The calculation of the *steady-state* responses in Eq. (6) requires first summing the contributions from the lateral connections corresponding to a set of neurons in a surrounding patch $n$, and then multiplying the contributions from each such patch with the classical RF response. However, if the contributions from the surround are sufficiently small, the computation becomes simpler and Eq. (6) for the firing rate of the neuron simplifies to

$$f_{j,x}^m = \frac{1}{\mathcal{N}_x^m} fc_{j,x}^m \left( 1 + \sum_{n \neq m}^N \sum_k W_{jk}^{mn} fc_{k,x}^n \right) \qquad (8)$$

**Extension to multi-layer networks**

When applied to multi-layer networks (e.g. deep neural networks), we treat each feature map as containing units which respond to a given feature at a specific location within the image. For the first layer of the network (which sees the image as input), the learned lateral connections are captured by the derivations above. For deeper layers, we use the same formalism and set of assumptions, learning lateral connections between the hidden units based on their activations over a set of training images. During inference, we pass the real-valued activations modulated by the learned lateral connections onto the next layer (we do not perform any probabilistic sampling).

---

[2]In practice, we add a small constant $\epsilon$ to the sum on the left before normalizing. This is equivalent to a null feature for when no substantial contrast is present in patch $n$.

