# OpenReview forum: "Convolutional neural networks with extra-classical receptive fields"
_NeurIPS.cc/2019/Workshop/Neuro_AI — Real Neurons & Hidden Units @ NeurIPS 2019 Poster_

### Official Review · AnonReviewer2 · 2019-09-25
**An investigation of lateral weights for improving classification**

**Clarity:** 5

**Comment:**

While minor aspects of the paper could be improved (see other comments), the paper seems like a useful contribution to the workshop.

One additional thing that would help in strengthening the submission would be a discussion around other normalization schemes currently used in deep learning, e.g. batch normalization, layer normalization, local response normalization. Comparing and contrasting these ideas, even at a conceptual level, would be useful.

**Category:**

Neuro->AI

**Clarity Comment:**

The presentation is, overall, quite clear. Further details in the experimental set-up and weight derivation would be helpful in the final version.

**Evaluation:**

5: Excellent

**Importance:**

4: Very important

**Importance Comment:**

This paper shows how lateral connections can make neural networks more robust to noise (in the data) in the setting of classification. Although the results are not necessarily groundbreaking, they provide a useful demonstration of where these lateral weights could assist in modern deep learning architectures.

**Intersection:**

4: High

**Intersection Comment:**

The paper primarily uses lateral weights as a mechanism for improving robustness in neural network models. There are some comparisons with neuroscience data in Figure 1. While these findings may be helpful to a neuroscience audience, the paper seems primarily geared toward the machine learning community.

**Rigor Comment:**

Overall, the authors are fairly rigorous in their evaluation, performing a reasonable set of experiments that compare different schemes (lateral vs. non-lateral), noise levels, noisy types, and datasets. A couple things stick out somewhat: 1) from Table 1, it appears as though the authors use a deeper architecture for their model, which may confound their results, and 2) the authors claim that they observed sparsification of feature activations with their method, however this is not backed up empirically. I would be helpful to include these results in order to make this claim.

**Technical Rigor:**

4: Very convincing

---

### Official Review · AnonReviewer3 · 2019-09-26
**Hybrid approach of supervised training of CNN, unsupervised training of lateral connections, may improve generalization.**

**Clarity:** 3

**Category:**

Neuro->AI

**Clarity Comment:**

Comparison of the proposed lateral hebbian rule to previous work, e.g. neocognitron?
Lacks discussion


**Evaluation:**

3: Good

**Importance:**

3: Important

**Importance Comment:**

Interesting Neuro->AI proposal to add lateral connections like in cortex, and quantify potential functional role
Preliminary but interesting results on MNIST
Less convincing for CIFAR-10

**Intersection:**

4: High

**Intersection Comment:**

A good example of Neuro->AI and back.  Ideas from Neuro, functional interpretation from AI application.

**Rigor Comment:**

Statistical significance of improvements due to proposed method for CIFAR-10 needs further scrutiny.

**Technical Rigor:**

3: Convincing

---

### Official Review · AnonReviewer1 · 2019-09-26
**Not theoretically or technically compelling**

**Clarity:** 3

**Comment:**

This submission proposes introducing "extra-classical receptive field" layers into neural networks. These layers apply weights that integrate over a slightly wider area than the preceding convolutional layer. Although the idea of extraclassical receptive fields is potentially interesting, the "extraclassical receptive field" here is just a convolutional layer that is trained in an "unsupervised" way that is neither well-described nor well-justified. There are significant problems with the evaluation that make it difficult to draw meaningful conclusions about the performance of the method, but it does not appear to meaningfully improve performance or robustness on CIFAR-10.

Strengths:

The proposed idea seems to modestly improve robustness of a small network trained on MNIST to large magnitude noise perturbations, versus a baseline that isn't totally fair.

Weaknesses:

The theoretical justification for the proposed technique equates the activations in hidden layers of neural network to probabilities that a feature is present, which is nonsensical. The hidden layers of a standard neural network do not implement probabilistic inference, and only the output of the network can be directly treated as a probability distribution.

It is unclear how the weights of the proposed model are implemented in practice.

The parameter-matched baseline is chosen to be architecturally different from the CNNEx model, rather than the same model with weights trained by backpropagation.

The CIFAR-10 experimental setup is unconvincing. The baseline is worse than the best result reported in the 2009 paper that introduced the dataset, and it is unclear whether the results would generalize to networks trained for more epochs with more layers and more filters.

The proposed method does not appear to achieve meaningfully better performance than the baseline on CIFAR-10, and actually performs significantly worse than the baseline when weight decay is applied.

**Category:**

Neuro->AI

**Clarity Comment:**

The paper is generally readable, but there is one major missing detail: It is not clear to me how the "learning rule" in Eq. 2 is actually implemented. It doesn't seem like it's actually an iterative learning rule, since there is an expectation over images. At least in a typical ML training setup, it's intractable to perform a forward pass for all training images at each set. Assuming the training setup is typical, do the authors backpropagate through the covariance computation on a minibatch and use full-batch moments at test time, or do they use some kind of exponential moving averaging over training?

**Evaluation:**

1: Very poor

**Importance:**

2: Marginally important

**Importance Comment:**

The idea of adding some kind of lateral connectivity to a deep learning model has been explored previously (e.g. in AlexNet), but it's always interesting to see work along these lines. However, the rationale behind the existing approach doesn't seem to make much sense, and the evaluation is inadequate to show whether it works. I do not think there's much for future work to build upon in this submission.

**Intersection:**

3: Medium

**Intersection Comment:**

This paper involves a neuroscience-inspired idea applied to a neural network. However, the implementation of the general idea has little relationship to neuroscience, and it's difficult to link the results to any insights that would be useful to neuroscience. Thus, the relationship seems surface-level.

**Rigor Comment:**

The proposed "rule" does not make much sense. First, it's not actually a learning rule, since it's not an iterative update, and it's not clear how it's implemented in practice (and whether the authors backpropagate through the computations involved). Second, the derivation seems to rely on a strange idea about how neural networks work. It refers to "the probability of the coded feature," but this doesn't make much sense since the real-valued features are passed directly to the next layer and not sampled from. There also seem to be a massive number of assumptions involved, which are not justified and do not make any intuitive sense, e.g. "each patch provides independent information" and $p(F_k^2|im_x^1 \approx p(F_k^2)$.

MNIST is a toy task and methods that improve robustness on MNIST do not necessarily transfer to other datasets, but the evaluation here is mostly reasonable. The original network is only slightly worse than a LeNet-5 baseline. In this setting, CNNEx seems to marginally outperform the network without weight decay and dropout at high distortion levels, but the proposed method is less effective than weight decay. When the methods are combined, it looks like the parameter-matched CNN outperforms CNNEx at most distortion levels, but the proposed method still yields gains at larger distortion levels. However, it is not clear to me why the parameter-matched CNN baseline consists of adding more filters to the baseline CNN, rather than training the "extraclassical receptive field" layers by backpropagation.

On CIFAR-10, the baseline achieves around 60%. This result is much worse than the 84.4% result reported for a three convolutional layer network in the dropout paper. The network likely performs poorly because it has very few filters and was trained for only 10 epochs. It's hard to know whether gains in this rather unrealistic network would translate to a network with more filters, much less a modern image classification network. But also, there aren't really gains in this setting. The accuracy in noise without weight decay is <1% vs. the parameter-matched baseline, and there is no advantage to the proposed method over a CNN with weight decay.

**Technical Rigor:**

1: Not convincing

---

### Decision · Program_Chairs · 2019-10-02

Accept (Poster)